# The Sum of Plasma Fatty Acids iso16:0, iso17:0, *trans11*-18:1, *cis9, trans11*-CLA, and *cis6*-18:1 as Biomarker of Dairy Intake Established in an Intervention Study and Validated in the EPIC Cohort of Gipuzkoa

**DOI:** 10.3390/nu13020702

**Published:** 2021-02-22

**Authors:** Alaitz Berriozabalgoitia, Juan Carlos Ruiz de Gordoa, Mertxe de Renobales, Gustavo Amores, Luis Javier R. Barron, Pilar Amiano, Miren Dorronsoro, Zelai Perea, Mailo Virto

**Affiliations:** 1Lactiker Research Group, University of the Basque Country UPV/EHU, Paseo de la Universidad 7, 01006 Vitoria-Gasteiz, Spain; alaitz.berrozabalgoitia@ehu.eus (A.B.); juancarlos.ruizdegordoa@ehu.eus (J.C.R.d.G.); mertxe.derenobales@ehu.eus (M.d.R.); gustavo.amores@ehu.eus (G.A.); luisjavier.rbarron@ehu.eus (L.J.R.B.); 2Department of Biochemistry and Molecular Biology, University of the Basque Country (UPV/EHU), Paseo de la Universidad 7, 01006 Vitoria-Gasteiz, Spain; 3Department of Pharmacy and Food Sciences, Faculty of Pharmacy, University of the Basque Country (UPV/EHU), Paseo de la Universidad 7, 01006 Vitoria-Gasteiz, Spain; 4Public Health Division of Gipuzkoa-Biodonostia Research Institute, Basque Regional Health Department, 20013 San Sebastian, Spain; epicss-san@euskadi.eus (P.A.); epicss-san@euskadi.es (M.D.); 5Laboratory of Biochemistry, Basque Regional Health Department, Araba University Hospital, 01009 Vitoria-Gasteiz, Spain; zelai.pereasainz@osakidetza.eus

**Keywords:** dairy products, dairy fat, rumenic acid, vaccenic acid, petroselinic acid, iso fatty acids, biomarkers, plasma, atherogenicity indexes

## Abstract

The questioned reliability of 15:0, 17:0, and *trans9*-16:1 acids as biomarkers of dairy fat intake also questions the relationship between the intake of these products and their health effects. Two studies were conducted in the same geographical region. In an intervention study, volunteers followed a diet rich in dairy products followed by a diet without dairy products. Plasma and erythrocyte fatty acids (FA) were analyzed, and their correlations with dairy product intakes were tested. The FA biomarkers selected were validated in the Gipuzkoa cohort of the European Prospective Investigation into Cancer and Nutrition (EPIC) observational study. The correlation coefficients between plasma concentrations of iso16:0, iso17:0, *trans11*-18:1, *cis9, trans11*-18:2, and *cis6*-18:1 and the dairy fat ingested are similar in both studies, indicating that their concentration increases by 0.8 µmol/L per gram of dairy fat ingested. The biomarkers are positively related to plasma triglycerides (*r* = 0.324 and 0.204 in the intervention and observational studies, respectively) and total cholesterol (*r* = 0.459 and 0.382), but no correlation was found between the biomarkers and atherogenicity indexes. In conclusion, the sum of the plasma concentration of the selected FAs can be used as biomarkers of dairy product consumption. A linear relationship exists between their plasma concentrations and ruminant product intake. These biomarkers allow for obtaining consistent relationships between dairy intake and plasma biochemical parameters.

## 1. Introduction

Dairy products comprise a group of heterogeneous food products consisting primarily of milk, cheese, and yogurts from ruminants, mostly cows, sheep, and goats. They provide a large number of essential nutrients that could be of benefit for most people. However, their positive effect on human health has been questioned due mainly to their large content of saturated fatty acids (FA). The review by Thorning et al. [1] concludes that the available scientific evidence indicates that milk and dairy products contribute to fulfilling nutritional recommendations and that they may protect against chronic diseases. On the contrary, other authors state that, although milk and dairy products have been included in many dietary guidelines, their association with reduced risk of developing cardiovascular diseases (CVD) is still controversial [2].

The contradictory results of published research could be due to any or several of the following reasons: the great variability of the nutritional composition of dairy products, different dietary patterns and intake of these products in various regions and countries, and the use of dietary questionnaires to obtain data on intake of dairy products. These methods are inherently imprecise in calculating the intake of dairy products and the nutrients provided by the diet. In addition, food composition is highly variable. As a result, coefficients of variation obtained in epidemiological studies are quite high and the conclusions are not consistent. For these reasons, it is necessary to define reliable biomarkers to determine as objectively as possible the intake of dairy products.

More than 400 different FAs have been identified in bovine milk fat [3]. Some of them are unique because they are synthesized de novo by rumen bacteria. Bacteria in the rumen synthesize odd-chain FA 15:0 and 17:0, iso or anteiso FA, and trans FA, primarily vaccenic (*trans11*-18:1) and rumenic (*cis9, trans11*-18:2, conjugated linoleic acid (CLA)) acids. These FAs can reach the mammary gland, where they are incorporated into milk lipids [4]. Thus, in spite of the many different types of dairy products, they have a characteristic FA composition [3]. In a recent review, Pranger et al. [5] concluded that 14:0, 15:0, 17:0, and *trans9*-16:1 (trans-palmitoleic acid) are biomarkers of total dairy and dairy fat intake commonly used in the scientific literature, the more frequently used ones being 15:0 and 17:0. Nevertheless, their validity was recently questioned [6] because they are found in foods other than dairy products and because their synthesis by β-oxidation of longer chain FAs in human tissues was demonstrated [7]. Rumenic acid [8] was also recently proposed as a dairy intake biomarker.

In most scientific works, molar or weight percentages of fatty acids in plasma or serum were positively correlated with total dairy intake [5]. Very few studies report the proportionality relationship between the amount of ingested dairy products and the absolute concentration of biomarkers in the samples [8,9].

Therefore, we present an intervention study to determine the most appropriate biomarkers for dairy intake and to calculate the proportionality relationship between their plasma concentration and the amount of dairy products ingested. The calculated relationship was then validated in an observational study carried out in the same geographical region. Subsequently, the correlation between the concentration of biomarkers and plasma biochemical parameters was analyzed to demonstrate the reliability of the proposed biomarkers. It was hypothesized that suitable fatty acid biomarkers, present almost exclusively in ruminant fat and showing a proportional relationship with ingested dairy fat, will allow for obtaining consistent relationships between dairy intake and plasma biochemical parameters.

## 2. Materials and Methods

### 2.1. Intervention Study

The Ethics Committee of the University of the Basque Country approved the study. The study was carried out in May and June 2013 in the Basque Country (Spain). All participants signed the Informed Consent document prior to the start of the study. Subjects were 10 adult volunteers (8 women and 2 men) of 42.9 ± 11.7 years of age, workers in the Faculty of the Pharmacy of the University of the Basque Country, with no known pathologies. At the beginning of the study, subjects answered a validated food frequency questionnaire [10] to determine their habitual diet. They were asked to follow their habitual diet, adding at least 375 g of full fat dairy products and recording the amount of dairy products ingested every day for 18 days. The amount of dairy fat ingested with each type of food during the first period was calculated from food composition tables [11]. At the end of the 18 days and for another 18 days, the subjects were asked to continue with their habitual diet but with no dairy products.

Blood samples were collected after overnight fasting on day 1, day 18, and the last day of the study (day 36) in 5 mL vacuum tubes impregnated with EDTA. Plasma and erythrocytes were separated by centrifugation at 800 g for 30 min. After separation, the samples were kept frozen at −80 °C until analyzed.

### 2.2. Observational Study

The Gipuzkoa (Basque Country, Spain) European Prospective Investigation into Cancer and Nutrition (EPIC) cohort was established between 1992 and 1995 as part of the EPIC multicenter prospective study [12]. The cohort was comprised of 8417 participants (49.5% men and 50.5% women with an average 51 and 48 years of age, respectively). Participants with known pathologies, those with an energy consumption greater than or lower than 3 standard deviations and those that had not fasted for at least 8 h before extracting the blood sample were excluded for the present study. Individuals were divided into four identical groups according to dairy intake. Information on usual intake was collected by the diet history method validated previously [13].

### 2.3. Fatty Acids Analysis

Methylation and extraction of total FA in plasma and erythrocytes was accomplished in one step without prior extraction of fat following a modified method of Bondia-Pons [14,15]. Fatty acid methyl esters (FAMEs) were prepared from 200 µL of plasma by sequentially, adding 2.5 mL of 0.5 M sodium methoxide in methanol followed by 2.5 mL of 14% boron trifluoride in methanol. The resulting FAMEs were extracted by adding 1.0 mL n-hexane and were collected in a vial with anhydrous sodium sulphate. For erythrocytes, after adding sodium methoxide, the cell suspension was sonicated for 3 minutes in 10 s cycles.

FAME separation was achieved by gas chromatography on a 100 m CpSil88 capillary column (Varian Inc., Mississauga, ON, USA). The chromatograph (Agilent Technologies, Madrid, Spain) was equipped with an FID detector. FAMEs were identified by comparing the retention times of the obtained peaks with those of authentic standards. The internal standard method was used to quantify the amount of each compound, with undecanoic (11:0), tridecanoic (13:0), and nonadecanoic (19:0) acids as internal standards. The absolute FA concentration was expressed in µmol/L units.

### 2.4. Plasma Biochemical Parameters Analysis

In the intervention study, the biochemical parameters (total cholesterol (TC), triglycerides (TG), cholesterol in low density lipoproteins (LDL-C), and cholesterol in high density lipoproteins (HDL-C)) were determined using commercial enzymatic kits (BioSystems, Barcelona, Spain) based on the spectrophotometric quantification of quinonimine. Lipoproteins were determined by the method of Kelley and Kruski [16].

In the observational study, TC, TG, and HDL-C were determined by enzymatic methods using an analyzer management system (ARCHITECT 16,000, Abbott diagnostics, Green Oaks, IL, USA). LDL-C was calculated using the equation of Friedewald [17].

The TC/HDL-C ratio and atherogenic index of plasma (AIP) were calculated, and the latter was calculated according to the formula AIP = log (TG/HDL-C), with both TG and HDL-C expressed in molar concentration [18].

### 2.5. Statistical Analysis

Food intake data were expressed as grams per day (g/day). The FA levels in plasma and erythrocytes in both studies were expressed in micromole per liter (µmol/L). All analyses were performed using the IBM-SPSS statistical software, version 25 (IBM, Chicago, IL, USA). The Shapiro–Wilks test was used to verify the normal distribution of the data of FA concentration and biochemical parameters. In the intervention study, repeated measures one-way analysis of variance (ANOVA) was used to compare the averages of FA concentrations and biochemical parameters obtained at baseline (day 1) and after completion of both diets (days 18 and 36) using the Bonferroni test for multiple comparison between days. Pearson correlations were used to determine the relationships between FA concentrations in the plasma and erythrocytes for each FA. A principal component analysis (PCA) including the concentrations of all FA and food intake parameters from dietary questionnaires was performed to identify FA related with the intake of various foods. A stepwise multiple linear regression analysis (MLRA) was done to determine the existence of a significant relationship between FA concentrations and dairy product intake. FA concentrations in plasma and erythrocytes were considered dependent variables, with the following being considered as independent variables: daily intake of dairy products or dairy fat (g/day); BMI; daily total energy intake (Kcal/day); and daily intake of other foods (g/day) such as olive oil, fish, red meat, and biscuits and cakes.

In the observational study, a descriptive statistical analysis and a one-way ANOVA with a Tukey post hoc test was conducted to determine if there were statistically significant differences in the FA concentrations and biochemical data among quartiles. PCA and MLRA were performed using the same factors as described for the intervention study. To analyze the linear relationships, individuals of the observational study were classified in deciles, with decile 0 including all cases corresponding to participants who declared no intake of dairy products. Least square averages of FA concentrations for each decile adjusted to total energy intake (Kcal/day) were calculated and graphed versus the median values of dairy products (or dairy fat) intake of each decile.

Partial correlations were analyzed between the biochemical parameter values and total dairy intake (g/day), dairy fat intake (g/day), and plasma concentration of the sum of FAs chosen as biomarker adjusted by BMI and total energy intake (Kcal/day). Statistical significance was declared at *p* < 0.05.

## 3. Results

### 3.1. Intervention Study

Participants in the intervention study (*n* = 10) were asked to follow their habitual diet (the baseline characteristics of individuals and their habitual diet are shown in Appendix A). During the intervention, in the period of dairy intake, the minimum amount of dairy products consumed by each participant was defined considering the intake of these products in the Spanish population based on a Mediterranean diet [19]. Each participant was free to choose product(s) and amount(s) of dairy products to consume according to her/his preferences and eating habits. Table 1 shows the average amount of dairy products ingested every day and the amount of dairy fat ingested with each type of food, calculated according to food composition tables [11]. The final amount consumed by participants was 544.3 ± 149.7 g/day (33.1 ± 9.2 g of dairy fat/day), which was higher than their habitual dairy intake (382.2 ± 123.2 g/day, Appendix A) and the normal dairy intake in the region [19].

Table 2 shows the concentration (µmol/L) of FAs in the plasma of participants. Comparing mean values with no dairy intake (NDD) and dairy-rich diet (DRD), it can be observed that data from the plasma after DRD are between 333% (petroselinic acid, cis6-18:1) and 10% (12:0) higher for 17 of the 43 individual FAs identified and quantified in the plasma. In the same way, the concentration of the sum of even-chain and odd-chain saturated FAs and the sum of CLAs are higher in the plasma of participants after the DRD. Likewise, in erythrocytes (Appendix A), significant increases (between 115% (rumenic acid) and 18% (15:0)) were observed in 21 individual FAs and in the sums of even-chain and odd-chain saturated FAs, and monounsaturated (*cis*- and *trans*-) and polyunsaturated FAs (CLA included). The concentrations of 10 FAs was significantly higher both in plasma and in erythrocytes after the DRD. As expected, most of the FA showing increased amounts were characteristic of dairy products: medium-chain FAs (10:0–14:0), 17:0, branched-chain FAs (iso and anteiso), vaccenic acid, and CLA (rumenic and *trans9, trans11*–18:2). To determine whether changes in FA concentration in plasma correlated with changes in erythrocytes, Pearson correlations between the concentration of each FA in plasma and erythrocytes (Table 2) were calculated. Overall, there is a high correlation between both fractions, with higher correlation values for those FAs showing increased concentrations with dairy intake (0.741 for 14:0, 0.613 for 17:0, 0.694 for vaccenic acid, and 0.821 for rumenic acid).

The PCA analysis carried out including plasma FA concentrations yielded ten PCs with eigenvalues higher than 1, explaining 88.05% of the variance. PC1 (32.85% of the total variance) includes 12 FAs, 11 of which (with loadings close to 0.9) are usually correlated with dairy consumption, such as 14:0, vaccenic, or rumenic acids. In addition, petroselinic acid was included in PC1 (loading 0.873). Dairy fat intake was also included in PC1, with a loading of 0.914. The results for erythrocyte samples were very similar, yielding 11 PCs, which explained 92.62% of the variance. PC1 explained 35.57% of the variance. Among the FAs with highest loadings included in PC1 were vaccenic (0.891), rumenic (0.818), 14:0 (0.816), and petroselinic (0.837) acids. PC1 also included dairy fat intake (0.933). Figure 1 shows the plasma (a) and erythrocyte (b) sample scores distributed in the two-dimensional coordinate system defined by PC1 and PC2. In both cases, PC1 allowed the separation of samples according to intake, or no intake, of dairy products. In the plasma samples (Figure 1a), PC2 included cis-monounsaturated FAs (present in the olive oil consumed in Mediterranean diets) with high weights and it allowed for the separation of samples (particularly in the dairy-free group) according to olive oil consumption.

A stepwise MLRA was applied including all FAs included in the PC1 as dependent variables and the following as independent variables: dairy intake (g/day) or dairy fat intake (g/day), BMI, energy consumption (Kcal/day), and intake (g/day) of foods that could be confusing factors (red meat, fish, olive oil, and cakes and biscuits). The highest values of the determination coefficients (R2) were obtained for the relationship between the concentration (μmol/L) of plasma FA and dairy fat intake (g/day) (Table 3). The analysis included the intake of olive oil and cakes and biscuits, eliminating the rest of the independent variables due to their lack of significance (Table 3 and Model 1). The determination coefficients are slightly lower for the concentration of erythrocyte FAs, with olive oil as the sole confusing food (Table 3 and Model 2). The coefficients improve when the analysis is conducted with respect to the sum of the concentrations of some FAs. Thus, the best determination coefficient in plasma is obtained with the sum of iso16:0, iso17:0, vaccenic, rumenic, and petroselinic (R^2^ = 0.871) and in erythrocytes with the sum of iso17:0, vaccenic, rumenic, and petroselinic acids (R^2^ = 0.774). The regression coefficients (B) were determined after subtracting the effect of significant independent variables (Table 3). B values for plasma and erythrocyte concentration (µmol/L) of the aforementioned sum of fatty acids and the dairy fat ingested (g/day) are 0.827 and 0.301, respectively (R^2^ = 0.849 and 0.761). Both the results of the PCA and the high R^2^ values obtained as well as those of the standardized regression coefficients (not shown) indicate that dairy fat intake is the main factor that determines the concentrations of these FA in plasma and erythrocytes.

### 3.2. Observational Study

A total of 160 randomized individuals selected from the Gipuzkoa cohort of the observational EPIC study [12] were divided into four equal groups according to their consumption of dairy products. Of the 160 participants selected, nine gave extreme values for at least one of the FAs previously selected as biomarkers of dairy fat intake. The final number of participants and the dairy intake of each quartile are shown in Table 4. Appendix A summarizes the participant’s baseline anthropometric and diet characteristics. All quartiles had comparable average values for the variables age, BMI, and total energy intake.

Of the 43 FAs identified in plasma samples, the concentrations of 9 FAs showed significant differences according to dairy intake (Table 4). Similar to that in the intervention study, branched-chain, odd-chain, and vaccenic FA concentrations increase as dairy intake increases. Moreover, significant differences were obtained for the concentrations of 15:0 and trans-palmitoleic acids but not for the concentration of rumenic acid. In most cases, significant differences are obtained between the two extreme quartiles, with intermediate values for quartiles 1 and 2.

The PCA distributed individual FA concentrations among 7 PCs, which explained 77.65% of the total variance. PC1 (32.2% of the variance) included FA related to dairy intake, with the highest loading for iso17:0 (0.885) and vaccenic acid (0.834). Figure 2 shows the distribution of the plasma sample’s scores for quartiles 0 and 3 in the coordinate system defined by PC1 and PC2. Although the sample scores are more dispersed than those in the intervention study, the samples are distributed along PC1 according to the intake of dairy products.

The stepwise MLRA showed that there is a significant correlation between the concentration of FAs and the intake of dairy products (g/day) (Table 3, Model 3, and R^2^ for the sum of FA = 0.131, *p* < 0.001). Of all the independent variables included, only the daily energy intake showed a significant effect in addition to dairy intake. The regression coefficient (B), determined after subtracting the effect of energy intake, of the aforementioned sum of FA and the dairy fat ingested (g/day) is 0.335 (R^2^ = 0.131, *p* < 0.001). Figure 3 shows a linear relationship between the sum of FA concentrations and the intake of dairy products and dairy fat. Similar correlations were obtained for all individual FA that showed a significant increase in their concentrations in the observational study (Table 3). 

### 3.3. Comparative Analysis

Overall, the concentrations of FAs selected as biomarkers are higher in the observational study than in the intervention study. The sum of concentrations of selected FA in quartile 0 (participants who declare no intake of dairy products) was 50.35 ± 16.77 µmol/L, whereas it was 19.56 ± 5.42 µmol/L in the intervention study after the NDD. Comparing quartile 3 of the observational study with the intervention study after DRD, the concentrations are higher in the observational study, in spite of the intake of dairy in quartile 3 (504.5 ± 134.5 g/day) being similar to that of the intervention study (544.3 ± 149.7 g/day). However, the increase in the concentration with increasing dairy intake in the former study was less than that in the latter study, as indicated by the B values (0.335 vs. 0.827 µmol sum FA.day/L.g milk fat).

One remarkable difference in the diet of both studies was the intake of red meat. In participants in quartile 0, it was over two-fold (87.8 ± 59.1 g red meat/day) that of participants in the intervention study (34.8 ± 21.8 g/day) (Appendix A). Therefore, the difference in the sum of selected FA concentrations might be due to the differences in red meat (ruminant-derived almost exclusively) intake. Furthermore, while red meat intake does not change during the intervention study, in the observational study, the intake of dairy products increases across the quartiles the intake of red meat decreases (from 87.8 ± 59.1 to 47.9 ± 32.2 g/day). Thus, the contribution of red meat to the concentrations of selected plasma FAs can be different in each quartile. To correct (in an approximate manner) for the likely contribution of red meat, the ratio “FA concentration/red meat intake” was calculated for quartile 0, assuming that, in this quartile, red meat is the only source of these FAs. The same ratio was calculated for the selected FA concentrations in the dairy-free diet in the intervention study. The calculated values are similar (Appendix A) in both studies (0.640 ± 0.360 and 0.718 ± 0.408 for the sum of selected FA in the intervention and observational studies, respectively). 

After subtracting the contribution of red meat from the concentrations of plasma FA in quartiles 1–3 of the observational study (Table 3 and Model 4), the determination coefficients improve (0.260 vs. 0.131 for the sum of FA) and regression coefficients (B) for the selected FAs in the observational study turn out to be similar to those obtained in the intervention study. Thus, the proportionality relationship between the sum of plasma concentrations of selected FAs and dairy fat intake is 0.8 µmol.day/L.g, approximately, in both studies.

### 3.4. Biochemical Parameters

The plasma lipid parameters for both studies are summarized in Appendix A. No statistically significant (*p* ≥ 0.05) differences were observed in either study when the mean values of the parameters were compared among the groups according to their dairy intake, as declared in the diet questionnaires. Nevertheless, in the analysis of partial correlations adjusted for BMI and energy intake (Kcal/day), significant correlations were found when plasma lipid parameters were correlated with the concentrations of the selected biomarkers (Table 5). In both studies, plasma concentrations of biomarkers are positively correlated with plasma TG, TC, LDL-C, and HDL-C, but no significant correlation was found with atherogenic indexes (TC/HDL-C or AIP).

## 4. Discussions

The present investigation compares the results obtained in an intervention study with data obtained in an observational study carried out in the same geographical region. No similar studies have been found in the scientific literature.

The overwhelming majority of the large number of published articles describing various FA as biomarkers of dairy consumption express FA concentrations as weight or molar percent. Those data only reflect the relative intake of each FA. Absolute concentrations yield objective quantitative information, without being influenced by the intake of other foods that may contain significant amounts of other FA [20].

Dietary intervention with high amounts of dairy products yields characteristic FA profiles of plasma and erythrocytes, which are substantially different from those of individuals with a low or null dairy intake. In addition, the inclusion of dairy fat increases data variability, as can be seen in Figure 1 and Figure 2, in accordance with de Oliveira Otto et al. [21], who found that the addition of daily servings of regular cheese, whole-fat milk, and butter were associated with SD increases in FA concentrations.

Few intervention studies [22,23,24,25] analyzed the effect of consuming controlled amounts of dairy products in the FA profile of various blood fractions. In the present study, 18 days are sufficient to observe changes in the plasma FA profile, which are highly correlated with changes in the erythrocyte FA profile, as previously described [9,24,26,27]. Erythrocytes constantly exchange FA in their membrane phospholipids with those of plasma phospholipids [27], although their half-life is 28 days. Thus, erythrocyte FAs are no better indicators of long-term changes than plasma FA, as suggested by other authors [5,26]. In addition, other authors [28,29,30] demonstrated that total serum FAs reflect FA intake better than lipid fractions such as cholesterol esters, phospholipids, or triglycerides. It has also been demonstrated that, when the diet does not change, plasma FA can be as good biomarkers of long-term intake as those in adipose tissue [28,31].

Although 15:0 [9,21,32,33] or both 15:0 and 17:0 [25,31,34] are the most used biomarkers of dairy intake, in the present intervention study, only 17:0 increased significantly. These differences among studies could be due to the fact that the concentrations of these FA in dairy products may differ in different countries [9]. In addition, their value as biomarkers has been recently questioned [6,7]. Some authors [9,34] have proposed *trans-*palmitoleic acid as a biomarker, although human tissues can also synthesize it from vaccenic acid [35]. Mayneris-Perxach et al. [36] found a significant correlation (*r* = 0.288) between dairy intake and the plasma level of rumenic acid; Zong et al. [37] found a significant correlation (*r* = 0.37) between the percent of erythrocyte *trans*-18:1 isomers and dairy intake; and Pranger et al. [8] proposed that the percent of rumenic and vaccenic acids in TG and phospholipids could be used as biomarkers of dairy consumption. In the present work, branched chain FA iso16:0 and iso17:0 [4], and vaccenic and rumenic FA [3] were proposed as biomarkers as there is no doubt about their rumen origin. In addition, petroselinic acid has not been related to dairy products intake although it has been described as intermediate in the rumen biohydrogenation of 18 carbon polyunsaturated FA [38] and it has been found in some cheeses [39].

In addition to its unequivocal origin, a biomarker must exhibit a proportionality relationship that allows for calculating the amount of food consumed through its plasma concentration. The present study demonstrates that the concentrations of the selected FA, both in plasma and erythrocytes, increased significantly as the amount of consumed dairy products increased. Because the absolute concentrations of these FA are quite low, the sum of their concentrations (Table 3) is proposed as a biomarker. Similarly, Pranger et al. [8] also proposed the sum of the molar percentages of 15:0, 17:0, vaccenic, and rumenic acids as biomarkers. Only one study [9] was found that described a proportionality relationship between ingested dairy products, and plasma and erythrocyte 17:0 and *trans*-palmitoleic concentrations. Aslibekyan et al. [40] demonstrated that the weight percentage of 15:0 and 17:0 in adipose tissue increases in a proportional manner as their consumption increases.

The results of the intervention study are comparable to those of the EPIC-Gipuzkoa cohort [12], within the same geographical region but twenty years apart. Both studies followed a very similar Mediterranean diet, with high intake of fruits, vegetables, cereals, legumes, fish, and olive oil and a moderate intake of eggs, meat, and dairy. The main difference is that the consumption of red meat and meat products is considerably lower in the intervention study. According to Garcia-Closas et al. [41], in Spain, during the 1990s, consumption of meat products was about 200 g/day, a value that is similar to that declared by participants in the observational study. In 2011, the consumption of meat and meat products was 145 g/day and that of red meat was 53.4 g/day [42]. These values are higher than those declared by participants in the intervention study, coinciding with the decreasing trend, perhaps due to the warnings relating red meat consumption and an increased risk of developing cancer [43].

Hjaståker et al. [44] determined the consumption of dairy products in the various EPIC cohorts. Both the Spanish and the Gipuzkoa cohorts gave values higher than other European cohorts, with 399 g/day for women and 332 g/day for men. According to Valera-Moreiras [19], the average habitual consumption of dairy products in Spain is 356 g/day. Thus, the consumption of dairy products within the population of this geographical region did not change appreciably in the period of time covered by the present study. Nebwy et al. [45] also demonstrated that dietary habits within a population change very little with time.

Although small differences in diet are reflected in plasma concentrations of FA, this study demonstrates that the concentration of the proposed biomarkers is mainly due to the sum of the contributions of dairy products and red meat and that the contribution of both types of food is similar in both studies. Several studies mention the contribution of these foods to the concentrations of biomarker FA, but no study has been found in which the contribution of each food is estimated. It has been determined that, for each gram of daily-consumed dairy fat, the sum of the concentrations of biomarker FA increases by approximately 0.8 µmol/L. However, the plasma total concentration of these FA depends on the intake of dairy fat as well as on other foods of ruminant origin. That is to say, the amount of ingested dairy products cannot be directly inferred from the plasma concentrations of these FA without knowing the intake of those other foods. In addition, given that habitual consumption of dairy products and red meat as well as the concentrations of the selected FA in these foods changes in various countries [9,43,45], it is not possible to extrapolate the results obtained in the present study to other populations without carrying out the same analysis. Saadatian-Elahi et al. [46] compared the plasma FA profiles of EPIC participants from different European regions and demonstrated that the factor “region” determines the variations observed among the different cohorts, which in turn, depend most likely on the different dietary habits and life styles in each region.

Finally, the present work shows that biomarkers yield solid and consistent correlations between the intake of dairy products and disease risk biomarkers. On the contrary, when dairy intake is calculated from dietary surveys, the results found in the literature are contradictory [47]. Thus, the present work shows that biomarker concentration is positively correlated with an increase in LDL-C and HDL-C but not in TC/HDL-C or AIP. Huth and Park [48] also concluded that the results of short-term intervention studies with high intake of full-fat dairy increased LDL-C and HDL-C but did not increase the ratio TC/HDL-C. The increase in the concentration of biomarkers FA was also correlated with an increase in plasma TG, although the levels were within the recommended range. These results are repeated in the two studies, in spite of their different design and the time span between the two studies, which attests to their validity. Patel et al. [49] also found that FAs measured in plasma and erythrocytes were more strongly associated with diabetes incidence than data obtained from food frequency questionnaires, emphasizing the importance of using objective biomarkers to determine intake.

The major limitation of the present study is the small sample size and the unbalanced composition of the intervention study. Nevertheless, in our opinion, the results obtained in the observational study validate the results obtained in the intervention study despite its limitations.

## 5. Conclusions

Two studies were carried out in the same geographical region with basic Mediterranean diet over a 20-year interval. The use of plasma concentrations of selected FAs (iso16:0, iso17:0, vaccenic, rumenic, and petroselinic acids) as biomarkers of dairy intake in an intervention study was validated in an observational study. The plasma concentration of the sum of these FAs increases proportionally as the intake of dairy products increases, and the relationship was stable over time. Dairy products are not the only food source of those FAs, and therefore, the intake cannot be inferred from their absolute plasma concentrations, although they can be used in comparative studies within a region, particularly when the consumption of red meat is known. These biomarkers allow for obtaining coherent relationships between food intake and plasma biochemical parameters. Dairy fat intake is positively correlated with an increase in TG, LDL-C, and HDL-C but not with TC/HDL-C or AIP.

## Figures and Tables

**Figure 1 nutrients-13-00702-f001:**
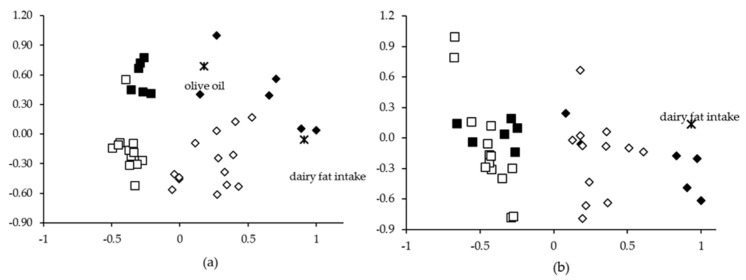
Two-dimensional plots representing the normalized scores for PC1 (x axis) and PC2 (y axis) obtained for plasma (**a**) and erythrocyte (**b**) samples of the intervention study (*n =* 10): samples after a diet without dairy products (□, ■); samples after a diet rich in dairy products (⬦, ⬥). Open symbols: olive oil consumption below 25 g/day; full symbols: olive oil consumption above 25 g/day. The principal component analyses (PCA) were carried out including plasma or erythrocyte concentrations of all individual FAs and the intake of dairy fat, olive oil, red meat, fish, and cakes and biscuits. PC1 explained 32.85% and 35.57% of the total variance in plasma and erythrocytes samples, respectively, and PC2 explained 13.25% and 12.18%, respectively.

**Figure 2 nutrients-13-00702-f002:**
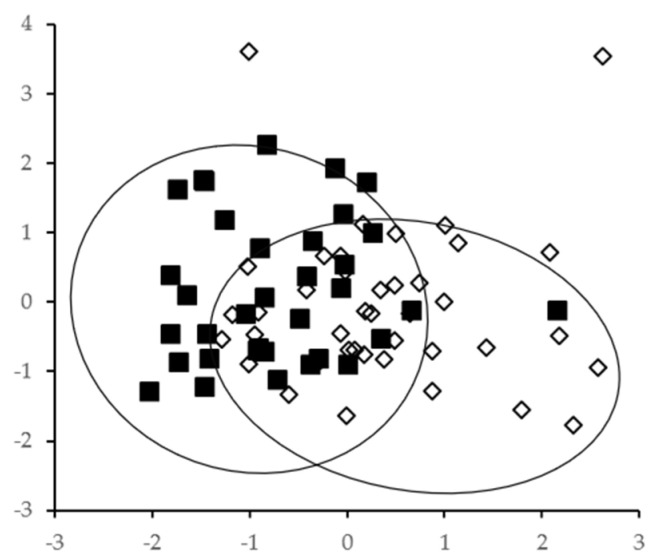
Two-dimensional plot representing the scores for PC1 (x axis) and PC2 (y axis) obtained for plasma samples of the observational study (*n* = 151): samples of quartile 0 (low dairy intake, ■); samples of quartile 3 (high dairy intake, ⬦). The principal component analysis (PCA) was carried out including plasma concentrations of all individual FAs and the intake of dairy fat, olive oil, red meat, fish, and cakes and biscuits. PC1 explained 32.2% of the total variance, and PC2 explained 14.1%.

**Figure 3 nutrients-13-00702-f003:**
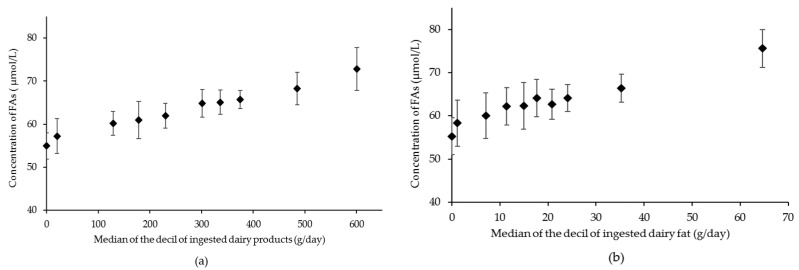
Relation between dairy products (**a**) or dairy fat (**b**) intake, and the sum of the concentrations of iso16:0, iso17:0, *cis6*-18:1, *trans11*-18:1, and *cis9,trans11*-CLA in plasma. Intake data were divided in deciles. The 0 decile includes all data from participants who declared no dairy intake. Fatty acid concentrations are least-squares means of each decile, adjusted for energy intake (Kcal/day). Bars represent ± SD values. Median values of intake for each decile are entered as a continuous variable to obtain the P value for the linear trend. *p*-values were < 0.001 in both cases. CLA, conjugated linoleic acid.

**Table 1 nutrients-13-00702-t001:** Intake of dairy products and fat from dairy products by participants in the intervention study.

Dairy Product	Intake, Mean (SD) ^1^	Dairy Fat Ingested, g/day ^2^
Whole milk	266.7 (140.5) mL/day	9.62 (5.23)
Full fat yogurt	182.9 (76.7) g/day	6.33 (2.68)
Fat cheese	55.9 (25.4) g/day	17.3 (8.74)
Others ^3^	38.9 (25.1) g/day	1.59 (1.46)
Total	544.3 (149.7) g/day	33.1 (9.22)

^1^ SD: Standard Deviation. ^2^ Calculated using food composition tables [11]. ^3^ Other milk products: cheesecake, curd, milk chocolate, béchamel, cream, ice cream, fresh cheese, and chocolate mousse.

**Table 2 nutrients-13-00702-t002:** Fatty acid (FA) concentration means (µmol/L) in the plasma samples from participants before (baseline) and after the intervention with the diet without dairy products (no dairy) and the diet with dairy products (with dairy), the standard error of the mean (SEM), and Pearson correlation between plasma and erythrocyte concentrations for each FA.

Fatty Acids	Baseline	No Dairy	With Dairy	SEM	*p* ^1^	Pearson Correlation ^2^
MCSFA						
10:0	27.87 ^b^	26.82 ^b^	32.88 ^a^	0.53	<0.001	0.145
12:0	94.08 ^b^	104.0 ^b^	113.4 ^a^	1.78	<0.001	0.155
14:0	99.83 ^b^	90.08 ^c^	153.1 ^a^	7.15	0.001	0.741 ***
15:0	36.66 ^a^	32.78 ^b^	34.40 ^a,b^	0.68	0.008	0.085
Sum	261.28 ^b^	253.6 ^b^	333.8 ^a^	1.68	<0.001	0.314 ***
LCSFA						
16:0	1996 ^a,b^	1868 ^b^	2231 ^a^	107.61	<0.001	0.377 **
17:0	21.02 ^b^	18.61 ^b^	24.53 ^a^	0.70	<0.001	0.613 ***
18:0	692.9	607.0	669.6	19.44	0.150	0.050
20:0	4.79	5.474	5.189	0.15	0.065	0.045
21:0	2.68	2.224	2.614	0.10	0.129	−0.058
22:0	13.86	13.92	17.39	1.41	0.356	0.022
24:0	48.25	42.26	51.82	6.07	0.230	−0.038
26:0	18.95 ^a^	16.39 ^b^	19.03 ^a^	0.49	0.001	0.396 **
Sum	2737 ^b^	2574 ^b^	3021 ^a^	127.30	0.001	0.234 **
ΣSFA	2930 ^b^	2827 ^b^	3355 ^a^	134.16	<0.001	0.240
ΣOCFA	60.36 ^a^	53.61 ^b^	61.59 ^a^	1.19	<0.001	0.146
BCSFA						
iso14:0	1.414	1.665	1.655	0.05	0.059	0.108
anteiso14:0	1.993	4.970	3.859	0.95	0.060	−0.024
iso15:0	3.275 ^b^	1.949 ^c^	4.555 ^a^	0.26	<0.001	−0.167
anteiso15:0	1.632 ^a^	0.7894 ^b^	2.278 ^a^	0.13	<0.001	0.192
iso16:0	5.599 ^b^	3.955 ^c^	6.867 ^a^	2.45	<0.001	−0.026
anteiso16:0	0.6568	0.5472	0.4586	0.03	0.068	
iso17:0	12.89 ^a^	6.495 ^b^	13.67 ^a^	0.73	<0.001	0.667 ***
anteiso17:0	33.95	33.49	30.84	1.22	0.247	0.345 **
Sum	61.42	56.06	66.98	3.05	0.073	−0.005
*c-*MUFA						
*cis9*-14:1	23.07 ^b^	15.98 ^c^	28.70 ^a^	1.05	<0.001	0.746 ***
*cis9*-16:1	129.1 ^a^	103.8 ^b^	128.8 ^a^	12.08	0.004	0.769 ***
*cis10*-17:1	11.40	10.48	10.65	0.26	0.048	0.179
*cis6*-18:1	2.353 ^b^	0.9266 ^c^	4.014 ^a^	0.40	<0.001	0.518 ***
*cis9*-18:1	1768	1788	1932	116.22	0.179	0.469 **
*cis11*-18:1	132.0 ^b^	145.4 ^a^	131.7 ^b^	4.26	0.019	0.461 **
*cis11*-20:1	11.41	10.70	10.33	0.58	0.478	0.286 *
*cis13*-22:1	120.5	88.84	118.8	4.92	0.688	0.521 **
Sum	2197	2182	2374	134.62	0.113	0.464 **
*t-*MUFA						
*trans10*-15:1	1.783 ^a,b^	1.496 ^b^	1.703 ^a^	0.05	0.041	0.265 *
*trans9*-16:1	2.243	1.827	2.256	0.12	0.179	0.178
*trans9*-18:1	16.78	20.34	17.92	1.13	0.139	0.117
*trans11*-18:1	5.422 ^b^	3.294 ^c^	7.533 ^a^	0.36	<0.001	0.694 ***
Sum	26.70	26.95	29.41	1.45	0.277	0.303 **
PUFA						
*trans*9, *trans12*-18:2	4.615	6.434	4.604	1.01	0.250	
*cis9*, *cis12*-18:2	3470	3231	3507	110.06	0.161	0.304 *
18:3 ω-6	161.1 ^a^	121.4 ^b^	135.0 ^a,b^	6.80	0.003	0.511 ***
18:3 ω-3	40.93 ^a,b^	32.53 ^b^	43.44 ^a^	2.32	0.014	0.679 ***
20:4 ω-6	945.4	904.9	923.1	25.72	0.420	0.346 **
20:5 ω-3	24.81 ^b^	29.15 ^a^	30.42 ^a^	0.69	<0.001	0.646 ***
22:6 ω-3	326.1	333.3	339.5	13.68	0.691	0.486 ***
Sum	4933	4659	4986	136.53	0.155	0.178
CLA						
*cis9*, *trans11*-CLA	10.74 ^a^	4.858 ^b^	13.34 ^a^	0.86	<0.001	0.821 ***
*trans10, cis12*-CLA	4.193	6.434	5.933	0.36	0.045	0.034
*cis9, cis11*-CLA	11.60	8.877	9.93	0.41	0.036	0.338 **
*trans9, trans11*-CLA	4.142 ^a,b^	2.886 ^b^	4.228 ^a^	0.25	0.005	0.387 **
Sum	30.70 ^a,b^	23.09 ^b^	33.43 ^a^	1.40	0.001	0.474 ***
TOTAL	10230 ^a,b^	9778 ^b^	10849 ^a^	391.62	0.017	0.221

^1^*p*-value < 0.05 was considered statistical significance. ^a,b,c^ letters indicate significant differences in compared groups. ^2,^* 0.01 < *p* < 0.05; ** 0.001 < *p* < 0.01; *** *p* < 0.001. MCFA, medium chain saturated fatty acids; LCSFA, long chain saturated fatty acids. ΣSFA, sum of saturated fatty acids; ΣOCSFA sum of odd numbered chain saturated fatty acids; BCFA, branched-chain fatty acids; c-MUFA, *cis*-monounsaturated fatty acids; t-MUFA, *trans-*monounsaturated fatty acids; PUFA, polyunsaturated fatty acids; CLA, conjugated linoleic acids.

**Table 3 nutrients-13-00702-t003:** Determination (R^2^) and regression (B) coefficients between the concentration (µmol/L) of fatty acids in different types of samples and total dairy fat intake (g/day) from whole milk, full fat yogurt, and fat cheese.

	Intervention Study						Observational Study (Plasma)			
	Plasma (Model 1)			Erythrocytes (Model 2)			Model 3		Model 4	
Fatty acid	R^2^	B ^a^	R^2 b^	R^2^	B ^a^	R^2 c^	R^2^	B ^a^	R^2^	B ^a^
14:0	0.849 ***	1.980	0.809	0.787 ***	0.401	0.755				
14:1	0.835 ***	0.383	0.802	0.598 ***	0.113	0.581				
15:0							0.130 ***	0.141	0.232 ***	0.277
iso15:0	0.495 ***	0.074	0.426							
anteiso15:0	0.729 ***	0.044	0.664	0.402 *	0.005	0.134				
iso16:0	0.769 ***	0.092	0.737				0.136 ***	0.042	0.259 ***	0.103
17:0	0.697 ***	0.205	0.605	0.423 **	0.097	0.424	0.140 ***	0.079	0.235 ***	0.242
iso17:0	0.793 ***	0.241	0.743	0.572 ***	0.109	0.530	0.134 ***	0.105	0.262 ***	0.246
*trans11*-18:1	0.771 ***	0.133	0.735	0.604 ***	0.043	0.593	0.115 ***	0.066	0.261 ***	0.121
*cis6*-18:1	0.712 ***	0.087	0.679	0.777 ***	0.049	0.777			0.141 **	0.037
*cis9, trans11*-CLA	0.804 ***	0.282	0.780	0.768 ***	0.097	0.752	0.062 **	0.099	0.201 ***	0.233
Sum ^d^	0.871 ***	0.827	0.849				0.131 ***	0.335	0.260 ***	0.740
Sum ^e^				0.774 ***	0.301	0.761				

Model 1: Multiple linear regression (fatty acid concentration (µmol/L) vs. total dairy fat intake (g/day) + olive oil intake (g/day) + cakes and biscuits intake (g/day)). Model 2: Multiple linear regression (fatty acid concentration in erythrocytes (µmol/L) vs. total dairy fat intake (g/day) + olive oil intake (g/day)). Model 3: Multiple linear regression (fatty acid concentration (µmol/L) vs. total dairy fat intake (g/day) + energy intake (Kcal/day)). Model 4: the same as Model 3 applied to data after resting the meat contribution (see text). ^a^ Regression coefficients for the variable “total dairy fat intake” according to each model. ^b^ Determination coefficients from fatty acid concentration vs. total dairy fat intake residuals graphics according to Model 1. ^c^ Determination coefficients from fatty acid concentration vs. total dairy fat intake residuals graphics according to Model 2. ^d^ ∑ *trans11*-18:1; *cis9, trans11*-CLA, iso16, iso17, and *cis6*-18:1. ^e^ ∑ *trans11*-18:1; *cis9, trans11*-CLA, iso17, and *cis6*-18:1. * 0.01 < *p* < 0.05, ** 0.001 < *p* < 0.01, *** *p* < 0.001.

**Table 4 nutrients-13-00702-t004:** Fatty acid concentration means (µmol/L) in plasma samples of participants in the observational study and the standard error of the mean (SEM).

	Dairy Consumption (g/day)					
	<0.71	9.37–217.74	221.43–351.25	>351.25		
Number	35	38	38	40		
Plasma fatty acids					SEM	*p* ^1^
15:0	17.92 ^c^	22.53 ^b^	22.61 ^b^	27.08 ^a^	0.63	<0.001
anteiso15	0.8840 ^b^	1.210 ^a^^,b^	1.191 ^a^^,b^	1.580 ^a^	0.07	0.012
iso16	7.265 ^b^	8.229 ^a^^,b^	8.216 ^a^^,b^	9.087 ^a^	0.26	0.005
iso17:0	17.66 ^b^	22.60 ^a^	23.67 ^a^	23.23 ^a^	0.56	<0.001
*trans9*-16:1	1.360 ^b^	1.637 ^b^	1.767 ^a^	1.702 ^a^^,b^	0.05	0.045
*cis9*-16:1	263.3 ^a^	216.7 ^a^^,b^	177.0 ^b^	168.2 ^b^	11.87	0.001
17:0	20.28 ^b^	23.08 ^a^^,b^	22.97 ^a^^,b^	24.67 ^a^	0.43	0.004
*trans11*-18:1	6.752 ^b^	9.658 ^a^^,b^	9.807 ^a^	10.84 ^a^	0.34	<0.001
*cis6*-18:1	2.050 ^b^	3.318 ^a^^,b^	3.369 ^a^^,b^	3.554 ^a^	0.17	0.008
*cis9, trans11*-CLA	16.82	20.52	20.63	20.50	0.59	0.066

^1^*p*-value < 0.05 was considered statistical significance. Different superscripted ^a,b,c^ letters indicate significant differences in compared groups. CLA, conjugated linoleic acid.

**Table 5 nutrients-13-00702-t005:** Partial correlations (r) between biochemical serum parameters and dairy for the samples taken in the intervention study and the observational study.

Biochemical Parameters	Intervention Study			Observational Study		
	r ^1^	r ^2^	r ^3^	r ^1^	r ^2^	r ^3^
TG, mg/dL	0.089	0.121	0.324 *	−0.069	−0.057	0.204 *
TC, mg/dL	0.064	0.189	0.459 **	−0.026	−0.003	0.382 **
LDL-C, mg/dL	0.139	0.203	0.348 *	0.008	0.009	0.328 **
HDL-C, mg/dL	−0.091	0.060	0.341 *	−0.050	0.004	0.206 *
TC/HDL-C	0.229	0.178	0.110	0.025	−0.019	0.112
AIP	0.247	0.179	0.207	−0.052	−0.045	0.007

^1^ Partial correlation between biochemical parameter and total dairy intake (g/day) adjusted for energy intake (Kcal/day) and BMI. ^2^ Partial correlation between biochemical parameter and total dairy fat intake (g/day) adjusted for energy intake (Kcal/day) and BMI. ^3^ Partial correlation between biochemical parameter and plasma fatty acid concentration (µmol/L of ∑iso16:0, iso17:0, *trans11*-18:1, *cis*9, *trans11*-CLA, and *cis6*-18:1) adjusted for energy intake (Kcal/day and BMI. *****
*p* < 0.05, ** *p* < 0.01. TG, triglycerides; TC, total cholesterol; LDL-C, cholesterol in low density lipoproteins; HDL-C, cholesterol in high density lipoproteins; AIP, Atherogenic Index of Plasma; log (TG/HDL-C).

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
