# Peer review of "The Sum of Plasma Fatty Acids iso16:0, iso17:0, trans11-18:1, cis9, trans11-CLA, and cis6-18:1 as Biomarker of Dairy Intake Established in an Intervention Study and Validated in the EPIC Cohort of Gipuzkoa"

_nutrients, 2021, doi:10.3390/nu13020702_

Round 1

Reviewer 1 Report

The study is extremely interesting and has great application in assessing dairy consumption. However, I have doubts about the selection of the intervention group.

  1. Why was the intervention group not matched better in gender and BMI to the group from the observational study? There were 80% women and 20% men in the intervention group compared to 45% men and 55% women in the observational study. BMI also differed significantly (22vs27.3)
  2. This resulted in a significantly lower mean energy intake in habitual diet of the intervention group compared to the observation study (1620.7 kcal vs 2245 kcal – 28%) and it also significantly affected the differences in consumption of different product groups. There were large differences not only in red meat or olive oil consumption but also in alcohol (35%), cakes and biscuits (32%) and fish consumption (17%).
  3. Consumption of these product groups may affect the lipid profile and also the correlation analysis of fatty acid concentration with plasma lipid parameters. Was the consumption of the remaining food groups carefully monitored during the intervention? Was the consumption different from habitual?

Thus, it is difficult to conclude that in terms of the association of biomarker levels with lipid profile, the intervention study confirms the results of the observational study.

In my opinion, the small size of the intervention group and its selection should be indicated as limitations of the study.

Author Response

- The study is extremely interesting and has great application in assessing dairy consumption. However, I have doubts about the selection of the intervention group.

Thank you very much for your kind comment.

- Why was the intervention group not matched better in gender and BMI to the group from the observational study? There were 80% women and 20% men in the intervention group compared to 45% men and 55% women in the observational study. BMI also differed significantly (22vs27.3)

The intervention study was designed as a preliminary exploratory work, so we decided to recruit participants among the staff of the Faculty of Pharmacy (University of the Basque Country).

The reason for the unbalanced sample is that the Faculty staff is also unbalanced. We cannot know exactly what the proportion of women was at the time of recruitment. Current data from the University of the Basque Country estimate 80% of women on the staff of the Faculties of Health Sciences. Similarly, in the work of Skeaff et al. (25) participants were recruited from students at the university. They recuited 20 people  (19 women and 1 man).

This resulted in a significantly lower mean energy intake in habitual diet of the intervention group compared to the observation study (1620.7 kcal vs 2245 kcal – 28%) and it also significantly affected the differences in consumption of different product groups. There were large differences not only in red meat or olive oil consumption but also in alcohol (35%), cakes and biscuits (32%) and fish consumption (17%).

Consumption of these product groups may affect the lipid profile and also the correlation analysis of fatty acid concentration with plasma lipid parameters. Was the consumption of the remaining food groups carefully monitored during the intervention? Was the consumption different from habitual?

The diet in the intervention study was the participants' habitual diet that was monitored by a FFQ.

Thus, it is difficult to conclude that in terms of the association of biomarker levels with lipid profile, the intervention study confirms the results of the observational study.

Differences found between both studies in diet and energy intake are most probably due to differences in methodology used for mesuring the habitual diet of participants. In the intervention study a FFQ was used; in the cohort study a dietary history questionnaire (DH) was designed to be used in Spain as part of a prospective European research project.

In any case, the statistical analysis was ajusted in all cases to energy intake and other food were included as posible confunders. And, as it is described in the manuscript, both the results of the PCA and the high R2 values obtained as well as those of the standardized regression coefficients (not shown) indicate that dairy fat intake is the main factor that determines the concentrations of these FA in plasma and erythrocytes.

We agree with the reviwer in the fact that a small part of these FA may come from other sources. However, our results show that there is a significant correlation between these fatty acids and some plasma parameters regardless of the source of these FA. And these correlation is similar in both studies.

In my opinion, the small size of the intervention group and its selection should be indicated as limitations of the study.

A paragraph stating this has been added at the end of the discussion.

Reviewer 2 Report

Mostly comments:

The experimental hypothesis should be included in “Introduction” section.

The manuscript should include a table on the correlation between selected FAs biomarkers and their sum and TC, LDL-C, HDL-C, TC/HDL-C and AIP.

In order to make the results presented in Tables 2 and 4 more readable, it would be better to give the standard error of the mean (SEM) instead of the standard deviation (SD) for particular groups. The applied method of presenting the results is correct (I have no substantive objections). It is only a suggestion.

Detailed comments:

Ln 15:  In addition to the chemical name, please add the symbol of FAs (C15:0, C17:0, trans-9-C16:1)

Ln 17-18: I suggest include the following information: Two studies were conducted in the same geographical region (intervention and observational). In an intervention study 10 volunteers followed a diet rich in dairy products 544.3 ± 149.7 g/day (33.1 ± 9.2 g of dairy fat/day)  followed by a diet without dairy products.

Ln 24-25: Since the abbreviations were used for the first time, an explanation should be added “TG (r = 0.324 and 0.204 in the intervention and observational study, respectively) and TC (r = 0.459 and 0.382)

Ln: 132: “Dairy food intake data are expressed as grams per day (g/d)” instead of “Food intake data are expressed as grams per day (g/d).”

Ln 127: “LDL-C was calculated using the equation of Friedewald” – literature reference should be provided.

Ln 221: The information “nd: under detection limit” is provided in the legend to Table 2, while it is missing in the table.

Ln 239: “The highest values of determination coefficients (R2) were obtained for the relationship between the concentration (μmole/L) of plasma FA and dairy fat intake (g/d)” instead of “Best determination coefficients (R2) were obtained for the relationship between the concentration (μmole/L) of plasma FA and dairy fat intake (g/d)”

Ln 243:  “Determination coefficients are slightly lower” instead of “Determination coefficients are somewhat lower”

Ln 276-278: Redraft this part of the text in a shorter form “A  total  of  160  randomized  individuals  from  the  Gipuzkoa  cohort  of  the  observational EPIC study [12] were selected. Selected individuals were divided into four equal groups according  to  their  consumption  of  dairy  products.  Of  the  160  participants,  nine  gave extreme  values  for  at  least one  of  the  FA  previously  selected as  biomarkers  of  dairy  fat intake.  Excluding  these  outliers,  the  final  number  of  participants  was  151.  Thus,  the resulting quartiles were: no dairy intake (< 0.71 g/d), low dairy intake (9.37 -217.74 g/d); moderate   dairy   intake   (221.43 -351.25   g/d)   and   high   dairy   intake   (>351.25   g/d).” 

Ln 283-289: The following information is not directly relevant to the study: “Supplementary Table 1 summarizes the participant’s baseline anthropometric and diet characteristics. All quartiles had comparable average values for the variables age, BMI and  total  energy  intake.  However,  consumption  of  certain  food  groups,  adjusted  for energy, was significantly different. There is an inverse relationship between dairy intake and that of red meat (partial correlation r = -0.350, P < 0.01), fish and fish products (r = -0.257, P < 0.01)  and  alcohol  (r = -0.458,  P < 0.01).  High  dairy  consumers  had  the  highest  intake  of cakes and biscuits (r = 0.245, P < 0.01).”, please add.

Ln 296-299: I would suggest placing this information in chapter 3.3. Comparative analysis: “Overall, comparing the concentrations of these FA in quartiles 0 and 3 with those obtained in the intervention study without and with dairy, respectively, the concentrations are higher in the observational study, in spite of the intake of dairy in quartile 3 (504.5 ±134.5 g/d) being similar to that of the intervention study (544.3 ± 149.7 g/day).”

Ln 324-325: I do not know where the R2 and P values came from: “Regression coefficient (B), determined after subtracting the effect of energy intake, of the aforementioned sum of FA and the dairy fat ingested (g/d) is 0.335 (R2 = 0.110, P < 0.001).”

Ln 347-349: Please rewrite this sentences for more clarity: “Intake data were divided in deciles, including in the 0 decile all data from participants who declared no dairy intake. Fatty acid concentrations are least-squares means of each decile, adjusted for energy intake (Kcal/d). Bars represent ±SD values. P values for linear trend were calculated by entering median values of each decile of intake into models as a continuous variable. P for trend was <0.001 for all.”

Ln 433-436: Please reedit in the short form: “The results of the intervention study are comparable to those of the Gipuzkoa cohort [12] of the EPIC observational study, both within the same geographical region. Twenty years separate both studies. Yet, participants in both studies followed a very similar Mediterranean diet with a relatively high consumption of fruits, vegetables, cereals, legumes, fish and olive oil and a moderate consumption of eggs, meat, dairy and meat products.”

References  - standard abbreviations for Journals in case of references  1 and 2 - Food Nutr Res, Clin Nutr

Throughout the paper, please pay attention to spaces (this is especially true when presenting values and units, significance of differences, e.g. ln: 95, 113, 175, 282, 323, 326, 338-339).

Author Response

The experimental hypothesis should be included in “Introduction” section.

The hypothesis was included at the end of the introduction

The manuscript should include a table on the correlation between selected FAs biomarkers and their sum and TC, LDL-C, HDL-C, TC/HDL-C and AIP.

Table 5 was included with these data

In order to make the results presented in Tables 2 and 4 more readable, it would be better to give the standard error of the mean (SEM) instead of the standard deviation (SD) for particular groups. The applied method of presenting the results is correct (I have no substantive objections). It is only a suggestion.

Done

Detailed comments:

Ln 15:  In addition to the chemical name, please add the symbol of FAs (C15:0, C17:0, trans-9-C16:1)

Done. Nevertheless, we used the abbrevation for FA without “C”, 15:0, 17:0…as appeared in other articles of the same journal.

Ln 17-18: I suggest include the following information: Two studies were conducted in the same geographical region (intervention and observational). In an intervention study 10 volunteers followed a diet rich in dairy products 544.3 ± 149.7 g/day (33.1 ± 9.2 g of dairy fat/day)  followed by a diet without dairy products.

The inclusion of this information, in addition to the abbreviations in the previous point, results in an abstract of more than 200 words, which is the limit according to the "Instructions for authors". Perhaps the editor can decide on this point.

Ln 24-25: Since the abbreviations were used for the first time, an explanation should be added “TG (r = 0.324 and 0.204 in the intervention and observational study, respectively) and TC (r = 0.459 and 0.382).

Done

Ln: 132: “Dairy food intake data are expressed as grams per day (g/d)” instead of “Food intake data are expressed as grams per day (g/d).”

This sentence does not only refer to dairy products. It also refers to other types of foods, which are listed in supplementary table 1, and which have been used in some of the statistical analyses.

Ln 127: “LDL-C was calculated using the equation of Friedewald” – literature reference should be provided.

Done

Ln 221: The information “nd: under detection limit” is provided in the legend to Table 2, while it is missing in the table.

It has been deleted.

Ln 239: “The highest values of determination coefficients (R2) were obtained for the relationship between the concentration (μmole/L) of plasma FA and dairy fat intake (g/d)” instead of “Best determination coefficients (R2) were obtained for the relationship between the concentration (μmole/L) of plasma FA and dairy fat intake (g/d)”

Done

Ln 243:  “Determination coefficients are slightly lower” instead of “Determination coefficients are somewhat lower”

Done

Ln 276-278: Redraft this part of the text in a shorter form “A  total  of  160  randomized  individuals  from  the  Gipuzkoa  cohort  of  the  observational EPIC study [12] were selected. Selected individuals were divided into four equal groups according  to  their  consumption  of  dairy  products.  Of  the  160  participants,  nine  gave extreme  values  for  at  least one  of  the  FA  previously  selected as  biomarkers  of  dairy  fat intake.  Excluding  these  outliers,  the  final  number  of  participants  was  151.  Thus,  the resulting quartiles were: no dairy intake (< 0.71 g/d), low dairy intake (9.37 -217.74 g/d); moderate   dairy   intake   (221.43 -351.25   g/d)   and   high   dairy   intake   (>351.25   g/d).” 

We have shortened it by removing data from the text and referencing the table where this data is collected.

Ln 283-289: The following information is not directly relevant to the study: “Supplementary Table 1 summarizes the participant’s baseline anthropometric and diet characteristics. All quartiles had comparable average values for the variables age, BMI and  total  energy  intake.  However,  consumption  of  certain  food  groups,  adjusted  for energy, was significantly different. There is an inverse relationship between dairy intake and that of red meat (partial correlation r = -0.350, P < 0.01), fish and fish products (r = -0.257, P < 0.01)  and  alcohol  (r = -0.458,  P < 0.01).  High  dairy  consumers  had  the  highest  intake  of cakes and biscuits (r = 0.245, P < 0.01).”, please add.

In our opinion, this type of study should refer to the baseline anthropometric and dietary characteristics of the participants, even if they are not relevant. Therefore, we think that we should keep at least the first part of the paragraph.

Ln 296-299: I would suggest placing this information in chapter 3.3. Comparative analysis: “Overall, comparing the concentrations of these FA in quartiles 0 and 3 with those obtained in the intervention study without and with dairy, respectively, the concentrations are higher in the observational study, in spite of the intake of dairy in quartile 3 (504.5 ±134.5 g/d) being similar to that of the intervention study (544.3 ± 149.7 g/day).”

Done

Ln 324-325: I do not know where the R2 and P values came from: “Regression coefficient (B), determined after subtracting the effect of energy intake, of the aforementioned sum of FA and the dairy fat ingested (g/d) is 0.335 (R2 = 0.110, P < 0.001).”

These values are shown in table 3, Model 3 (Multiple linear regression (fatty acid concentration (µmole/L) vs total dairy fat intake (g/ d)+ energy intake (Kcal/d) in the observational study.

Nevertheless, there is a mistake in the R2 value (it is 0.131 instead 0.110). The mistake has been corrected in the text.

Ln 347-349: Please rewrite this sentences for more clarity: “Intake data were divided in deciles, including in the 0 decile all data from participants who declared no dairy intake. Fatty acid concentrations are least-squares means of each decile, adjusted for energy intake (Kcal/d). Bars represent ±SD values. P values for linear trend were calculated by entering median values of each decile of intake into models as a continuous variable. P for trend was <0.001 for all.”

Done. We hope the sentence is clearer now than before.

Ln 433-436: Please reedit in the short form: “The results of the intervention study are comparable to those of the Gipuzkoa cohort [12] of the EPIC observational study, both within the same geographical region. Twenty years separate both studies. Yet, participants in both studies followed a very similar Mediterranean diet with a relatively high consumption of fruits, vegetables, cereals, legumes, fish and olive oil and a moderate consumption of eggs, meat, dairy and meat products.”

We have reduced it as much as we could

References  - standard abbreviations for Journals in case of references  1 and 2 - Food Nutr Res, Clin Nutr

Done

Throughout the paper, please pay attention to spaces (this is especially true when presenting values and units, significance of differences, e.g. ln: 95, 113, 175, 282, 323, 326, 338-339).

Done

Reviewer 3 Report

The study by Alaitz Berriozabalgoitia and co-authors is very well-written, and seems to be carefully performed and analyzed. It is certainly of interest to a wider community of nutrition researchers, since the reliability of biomarkers to assess the intake of a food group is highly relevant. A comprehensive list of fatty acids has been analyzed under dairy-free and high-dairy intake conditions and compared to a historical dataset from a population of the same area and with comparable dietary habits. The conclusions seem sound.

Major comment: While Figure 3 is a useful and illustrating figure (relationship between dairy intake and plasma values), the presentation of significant correlations of all fatty acids as table (Table 2) is less informative.

 The P-values for fatty acid concentrations often show high significance (P<0.001), while some of the corresponding correlation coefficients are rather weak. Some correlations are unexpectedly positive (or negative) - e.g. apparent decrease of anteiso17:0 concentration under high-dairy intake conditions gives a positive correlation. This makes it difficult to assess the quality of the correlations. The correlations should be provided as graphs for all +/- 50 fatty acids presented in Table 2 as supplementary data.

Very minor point: in the header of Table 3, it should read erythrocytes rather than erythrocites.

Author Response

The study by Alaitz Berriozabalgoitia and co-authors is very well-written, and seems to be carefully performed and analyzed. It is certainly of interest to a wider community of nutrition researchers, since the reliability of biomarkers to assess the intake of a food group is highly relevant. A comprehensive list of fatty acids has been analyzed under dairy-free and high-dairy intake conditions and compared to a historical dataset from a population of the same area and with comparable dietary habits. The conclusions seem sound.

Thank you very much for your kind comments

Major comment: While Figure 3 is a useful and illustrating figure (relationship between dairy intake and plasma values), the presentation of significant correlations of all fatty acids as table (Table 2) is less informative.

In our opinion, data in table 2 are of great relevance, as they are the basis of all the study. In addition, data of so many plasma fatty acids, expressed in concentration units (µmole/L), are very scarce in scientific literature. Therefore, they can be reference for future works.

 The P-values for fatty acid concentrations often show high significance (P<0.001), while some of the corresponding correlation coefficients are rather weak. Some correlations are unexpectedly positive (or negative) - e.g. apparent decrease of anteiso17:0 concentration under high-dairy intake conditions gives a positive correlation. This makes it difficult to assess the quality of the correlations. The correlations should be provided as graphs for all +/- 50 fatty acids presented in Table 2 as supplementary data.

Pearson correlation values in Table 2 are the correlations between plasma and erythrocyte concentration for each FA. To calculate them, we utilized all the data available for each FA in plasma and compared them with all the data available in red blood cells. Therefore, a positive correlation means that the concentration behavior is similar in both blood compartments. Therefore, they are not directly related to the significant differences found between values of the different diets groups within the plasma samples.

This is an important result because it indicates that it is not necessary to isolate erythrocytes to measure biomarkers. So, we propose to keep these data, as they are, in table 2.

Very minor point: in the header of Table 3, it should read erythrocytes rather than erythrocites.

Done